Maintaining fixation does not increase demands on working memory relative to free viewing

Armson Michael J. 1 2
Ryan Jennifer D. 1 2 3
Levine Brian blevine@research.baycrest.org 1 2 4
1 Rotman Research Institute, Baycrest Health Sciences , Toronto , Ontario , Canada
2 Department of Psychology, University of Toronto , Toronto , Ontario , Canada
3 Department of Psychiatry, University of Toronto , Toronto , Ontario , Canada
4 Department of Medicine (Neurology), University of Toronto , Toronto , Ontario , Canada
Martinez-Conde Susana
Electronic publication date: 2019 Apr 29
Publication date: 2019
Volume: 7
Electronic Location ID: e6839
Received 2018 Dec 9; Accepted 2019 Mar 22
Copyright: ©2019 Armson et al.
Copyright year: 2019
Copyright holder: Armson et al.
License: This is an open access article distributed under the terms of the Creative Commons Attribution License, which permits unrestricted use, distribution, reproduction and adaptation in any medium and for any purpose provided that it is properly attributed. For attribution, the original author(s), title, publication source (PeerJ) and either DOI or URL of the article must be cited.
License URL: https://creativecommons.org/licenses/by/4.0/

Keywords: Eye movements, Memory, Working memory, Long-term memory, Free viewing, Fixation, Finger tapping, N-back task, Visual processes, Imagery processes

Funding: Canadian Institutes of Health Research No. MOP-148940 This work was supported by the Canadian Institutes of Health Research (No. MOP-148940). The funders had no role in study design, data collection and analysis, decision to publish, or preparation of the manuscript.

==============================
The comparison of memory performance during free and fixed viewing conditions has been used to demonstrate the involvement of eye movements in memory encoding and retrieval, with stronger effects at encoding than retrieval. Relative to conditions of free viewing, participants generally show reduced memory performance following sustained fixation, suggesting that unrestricted eye movements benefit memory. However, the cognitive basis of the memory reduction during fixed viewing is uncertain, with possible mechanisms including disruption of visual-mnemonic and/or imagery processes with sustained fixation, or greater working memory demands required for fixed relative to free viewing. To investigate one possible mechanism for this reduction, we had participants perform a working memory task—an auditory n-back task—during free and fixed viewing, as well as a repetitive finger tapping condition, included to isolate the effects of motor interference independent of the oculomotor system. As expected, finger tapping significantly interfered with n-back performance relative to free viewing, as indexed by a decrease in accuracy and increase in response times. By contrast, there was no evidence that fixed viewing interfered with n-back performance relative to free viewing. Our findings failed to support a hypothesis of increased working memory load during fixation. They are consistent with the notion that fixation disrupts long-term memory performance through interference with visual processes.

Introduction

Eye movements have been shown to play a role in the encoding (Loftus, 1972; Henderson, Williams & Falk, 2005), maintenance (Olsen et al., 2014; Wynn et al., 2018), and retrieval (Richardson & Spivey, 2000; Johansson & Johansson, 2014) of memories. One study design that has been integral in demonstrating an involvement of eye movements in memory is a comparison of memory performance during free and fixed viewing. In this paradigm, participants are instructed to perform a memory task while either freely examining a computer screen during viewing, or while maintaining fixation on a central cross. The intention is to influence participants’ behaviour according to an ‘eye movement’ versus ‘no eye movement’ manipulation to investigate the effect of eye movements on memory performance. These effects are marked at encoding (Henderson, Williams & Falk, 2005): reductions on memory performance are observed following fixed, compared to free, viewing. Reductions in memory, though to a lesser degree (Damiano & Walther, 2019), have also been observed when fixed viewing is enforced at retrieval, including decreases in the retrieval of spatial localization of objects (Johansson & Johansson, 2014), recall of scenes (Johansson et al., 2012), and autobiographical memory (Lenoble, Janssen & Haj, 2018 see also El Haj et al., 2014; El Haj et al., 2017; El Haj & Lenoble, 2018).

While the effect of free versus fixed viewing on mnemonic performance is established, the mechanisms of this effect are uncertain. One set of hypothesis centres on the integration and/or recapitulation of visual details. Given the links between visual imagery and memory (Brewer & Pani, 1996; Rubin, Schrauf & Greenberg, 2003; Hassabis, Kumaran & Maguire, 2007), eye movements may reflect visual processes that are causally related to mnemonic operations. Eye movements may enable binding processes that occur along the visual processing hierarchy that serve to integrate information across space and time into a lasting memory representation (Liu et al., 2017). Additionally, eye movements reflect rehearsal and retrieval of previously studied information (Ryan et al., 2000; Olsen et al., 2014). Broadly speaking, these visual-mnemonic mechanisms are consistent with neuroanatomical data showing connectivity amongst oculomotor control regions, visual imagery regions, and the medial temporal lobes (Sheldon et al., 2016; Shen et al., 2016; Ryan et al., 2018). Fixation may diminish memory performance by disrupting such visual-mnemonic processes.

On the other hand, fixating on the central cross during fixed viewing can be considered as a secondary task that requires top-down control over one’s eye movements. Working memory tasks, such as non-visual mental arithmetic (Siegenthaler et al., 2014; Gao, Yan & Sun, 2015) or digit memorization (Dalmaso et al., 2017), have been shown to influence the rate of smaller, lower-velocity eye movements, called microsaccades, during fixation. Such controlled motor activity (Quinn & Ralston, 1986), as well as controlled eye movements specifically, have been shown to make demands on working memory capacity (Postle et al., 2006) and dorsolateral prefrontal cortical function (Munoz & Everling, 2004; Pierrot-Deseilligny et al., 2005) which could in turn disrupt task performance (Lenoble, Janssen & ElHaj, 2018). To the extent that fixation effects on memory can be accounted for by these general reductions in available resources, other explanations, including those involving visual processing, are confounded.

Support for either the visual processing or motor interference account of the free/fixed effect is indirect, derived from theory and from brain imaging studies. In order to adjudicate between these two accounts of the free/fixed effect, we conducted an experiment directly comparing the behavioural effects of fixation versus a simultaneous motor task.

To directly test the effects of top-down attentional interference from secondary motor tasks, we used an auditory n-back test of working memory (Kane et al., 2007; Jaeggi et al., 2010) as the primary task, enabling simultaneous monitoring of participants’ eye movements and verbal output in relation to an ongoing stream of information. Participants performed the auditory n-back task under free or fixed viewing conditions, or during a simultaneous finger tapping task known to be demanding of working memory (Quinn & Ralston, 1986; Moscovitch, 1994). As a secondary task, finger tapping has been shown to disrupt verbal memory (Friedman, Polson & Dafoe, 1988) and rehearsal (Hicks, Provenzano & Rybstein, 1975), similar to how eye movements would ostensibly interfere with verbal responses under a motor interference account (Brooks, 1968; Byrne, 1974). In addition to isolating motor interference independent of the oculomotor system, the inclusion of the finger tapping task allowed for proof of principle that the auditory n-back task is sensitive to the effects of interference from a secondary motor task. We hypothesized that accuracy would be lower in the finger tapping than in the free viewing condition, consistent with the known general resource limitation of the finger tapping task.

If the fixed viewing and finger tapping conditions both disrupted performance on the auditory n-back task relative to free viewing, this would suggest that controlled motor activity common to both conditions makes demands on working memory, supporting the hypothesis that the effects of free versus fixed eye movement manipulation can be attributable to general resource reduction from a simultaneous controlled motor task. On the other hand, a selective effect of finger tapping on auditory n-back performance would suggest that the oculomotor requirement to fixate does not induce working memory demands, supporting alternative accounts, such as those based on visual processing.

Materials & Methods

Participants

Twenty healthy young adults between the ages of 18 and 30 years old (13 females, mean age = 24.5 years, mean education = 17.3 years) were recruited through the Baycrest participant pool. Participants were ensured anonymity, and were informed of any risks of the experiment, as well as their right to withdraw at any time without consequence. The study was approved by the Baycrest Research Ethics Board (Approval Number: 02-01), and all the participants provided written informed consent before any experiments were completed. Exclusion criteria were history of neurological disease, active significant medical illness, serious psychiatric illness, or substance abuse during the last 12 months.

Participants reported their handedness as part of the screening questionnaire. Of our 20 participants, 17 were right-handed, two were left-handed, and one was ambidextrous. The finger tapping pattern was always performed with the right hand (described in more detail below), meaning that some participants performed the task with their dominant hand and others with their non-dominant hand. Right-handed finger tapping shows increased interference with verbal cognition in right-handed individuals (Kinsbourne & Hiscock, 1978). For this reason, handedness was included as a factor in our analysis of n-back performance across the motor conditions; it did not impact the results.

Procedure

Participants completed an auditory n-back task while simultaneously performing three dual task conditions. Participants heard spoken letters presented to them one after the other via earphones. Participants responded to the appropriate target letter by verbalizing the statement “yes” into a microphone. For the 0-back task, participants responded every time they heard the target letter “X”. For the 1-back task, participants responded when they heard a letter repeated one after the other (for example, “B-B”). For the 2-back task, participants responded for every letter that was the same as the letter presented two letters before (for example, “B-F-B”). During each of these tasks, participants were also presented with lure items, for which they were required to withhold a response. For a schematic outlining the n-back task, please see Fig. 1.

Figure 1 Auditory N-Back Task.

A schematic depicting our auditory 0-, 1-, and 2-back tasks. In all three tasks, each letter was presented over earphones, with a jittered interstimulus interval of 800-975 ms between letters.

Participants performed the auditory n-back task while viewing the screen according to three different dual task conditions. The two oculomotor conditions were (1) free viewing and (2) fixed viewing. In the free viewing condition, participants were instructed to move their eyes freely anywhere on the screen while completing the n-back task. In the fixed viewing condition, participants were instructed to fixate on a central cross at all times while responding to the n-back task. If their eyes strayed outside of a 400x400 pixel window surrounding the fixation cross, the cross would flash red to communicate that they should move their eyes back to the cross as quickly as possible. The fixed viewing condition was conducted in this manner after participants made considerable viewing errors during pilot testing with a smaller fixation window, resulting in increased distraction caused by the error feedback. To be sure that any effects observed during the fixed viewing condition were due to limited eye movements and not to processing of the error signal, we enlarged the viewing window slightly, so that eye movements would still be constrained relative to free viewing. For the finger tapping condition, participants could examine the screen freely, but were instructed to continuously and steadily execute a finger tapping pattern with their right hand on the keyboard in front of them. The pattern was ‘1’-‘3’-‘2’-‘enter’ (i.e., index-ring-middle-pinky) typed repeatedly on the number pad (Moscovitch, 1994). Participants practiced this pattern to a point of proficiency during practice trials, but there was no penalty for out-of-sequence taps during the test trials.

Participants completed three practice trials for each n-back load (0-, 1-, and 2-back) and viewing condition (free and fixed viewing, and finger tapping) to become familiar with the tasks, followed by the dual-task viewing/n-back task in three blocks such that all combinations of n-back (0- 1- and 2-back) and viewing (free, fixed, finger tapping) were equally represented across blocks. The order in which these viewing/n-back combinations were presented was counterbalanced across blocks. Each viewing/n-back trial was equivalent in terms of the number of letters presented—64 letters (including 16 target letters). The jittered interstimulus interval (i.e., the amount of time between the presentation of each letter) was 800-975 ms. This was also the amount of time participants had to make a response. Vocal “yes” responses were recorded using a Blue SnowBall iCE USB Microphone.

Equipment used during data acquisition

The eye tracking equipment used in this experiment was the same as that described in Ryan, Riggs & McQuiggan (2010).

Eye tracker

Eye movements were monitored using an EyeLink II system (SR Research Ltd; Mississauga, Ontario, Canada). This head-mounted, video-based eye tracker recorded eye position in X, Y-coordinate frame at a sampling rate of 500 Hz, with a spatial resolution of <0.1°. One camera was used to monitor head position by sending infrared markers to sensors placed on the four corners of the display monitor that was viewed by participants. Two additional cameras were mounted on the headband situated below each of the eyes, and infrared illuminators were used to note the pupil and corneal reflections. Eye position was either based on pupil and corneal reflections, or based on the pupil only. The padded headband of the eye tracker could be adjusted in two planes to comfortably fit the head size of an adult participant. Eyeglasses and contact lenses could be accommodated by the eye tracker.

Software

In this protocol, presentation of experimental stimuli and collection of eye position by the host PC were programmed through Experiment Builder, a software program specifically developed by SR Research Ltd to interface with the eye tracker host computer. Eye movement data were converted by the host computer to a series of fixation and saccade events that were time-locked to stimulus presentation. These fixation and saccade events were subsequently queried using Data Viewer, a software program developed by SR Research Ltd. Here, the detection of fixations and saccades were dependent on an online parser, which separated raw eye movement samples into meaningful states (saccades, blinks and fixations). If the velocity of two successive eye movement samples exceeded 30 degrees per second over a distance of 0.1°, the samples were labeled as a saccade. If the pupil was missing for three or more samples, the eye activity was marked as a blink within the data stream. Non-saccade and non-blink activity were considered fixations.

We used the experimental software package Presentation to program and present the auditory n-back task, and to record participant responses.

Data analysis

From the eye movement record, we derived measures of the number of fixations made to the computer screen for a given viewing/n-back trial, and the average saccade amplitude (ASA; the mean distance in degrees of visual angle of a participant’s saccade). To measure overall accuracy on the task, we computed d’ scores for each participant. These d’ scores were calculated as the standardized proportion of hits out of targets minus the standardized proportion of false alarms out of lure items (d′ = z (% hits) –z (% false alarms)). We also measured reaction times (RTs) to initiate an audible “yes” vocalization in registering these ‘hit’ responses. For each participant, the median RT on each condition was retained for further analysis.

We inspected the data from our eye movement and behavioural measures for outliers, defined as a data point greater than 2.5 standard deviations above or below the mean on a given condition. Two participants were outliers on multiple measures. One such participant was >2.5 standard deviations above the mean for RT on the free viewing 0-back condition, and >2.5 standard deviations below the mean for d’ on the free viewing and finger tapping 2-back conditions. The other multi-measure outlier was >2.5 standard deviations above the mean for number of fixations on the finger tapping 0-back condition, as well as ASA on each of the fixed viewing 0-, 1-, and 2-back conditions. These two multi-measure outliers were excluded from subsequent analyses. Of the remaining data, there were two participants who were outliers on RT—one on the fixed viewing 0-back and one on the finger tapping 2-back condition. Despite the above outliers, a test of skewness for each measure on each condition demonstrated that none of the distributions were significantly skewed, including the eight distributions with outliers. For this reason, the single measure outliers were retained and not adjusted for further analyses. To be sure that the retained outliers were not affecting the results, each analysis was performed both with and without these two single measure RT outliers. The retained outliers did not influence the pattern of results for any of the measures of interest.

For our data analysis, we ran both traditional frequentist and Bayesian models for each measure. First, we compared performance on the two eye movement measures (number of fixations and ASA) and two n-back measures (RT and d’) across the various n-back memory loads and viewing conditions, by running four 3x3 frequentist repeated measures ANOVAs. For each ANOVA, the three memory loads (0- versus 1- versus 2-back) and three viewing conditions (free viewing versus fixed viewing versus finger tapping) were included as within-subjects factors, with either number of fixations, ASA, RT, or d’ as the dependent variable. To determine whether participants performed the viewing task in adherence with instructions, we ran planned orthogonal contrasts of fixed viewing with each of the conditions allowing free examination of the screen—free viewing and finger tapping. For the n-back measures, we again ran linear planned contrasts for memory load, in this case as a manipulation check to see if participants would show the expected increase in RT and decrease in d’ scores with successive memory load conditions. For the eye movement measures, we ran linear planned contrasts to see if the number of fixations and ASA would decrease with increasing memory load, which could represent an eye movement response to increasing task difficulty (Meghanathan, Leeuwen & Nikolaev, 2015; Schut et al., 2017). In order to determine whether the motor interference effect of the secondary task was specific to finger tapping, we conducted hypothesis-driven planned orthogonal contrasts (free viewing versus finger tapping; fixed viewing versus finger tapping). For all four ANOVAs, significant interactions were probed using post hoc tests of simple effects, and effect size was measured as partial eta squared.

We complemented our frequentist analysis with four Bayesian repeated measures ANOVAs (Nathoo & Masson, 2016), composed of the same factors. These Bayesian models were run to obtain a Bayes factor for each measure—that is, a likelihood ratio providing a continuous measure of the observed data occurring under one hypothesis relative to another (Kass & Raftery, 1995). In terms of notation, BF10 represents the Bayes factor for the alternative over the null hypothesis, whereas BF01 gives the likelihood of the null over the alternative hypothesis. The Bayesian approach is especially useful for investigating null effects, which cannot be legitimately argued for within a frequentist framework (Wagenmakers, 2007). In this paper, we report both the frequentist and Bayesian results, as recommended by Dienes & Mclatchie (2018).

Results

Eye movements

To assess adherence to the viewing task instructions, we compared two eye movement measures—number of fixations and ASA—across the three viewing conditions and n-back loads. There was a main effect of viewing condition (F(2, 34) = 15.03, p < .001, ηp2=.469, Bayes factor BF10 = 2.17410) on the number of fixations executed. As expected, participants made more fixations during the two conditions involving free examination of the screen—free viewing (M = 162.96, SE = 13.04) and finger tapping (M = 143.22, SE = 9.09)—relative to the fixed viewing (M = 113.13, SE = 11.94) condition (vs. free viewing: F(1, 17) = 22.23, p < .001, ηp2=.567, BF10 = 6.3746; vs. finger tapping: F(1, 17) = 11.68, p = .003, ηp2=.407, BF10 = 5, 140.16). In visualizing the fixation data as heat maps, we could see that not only did participants make more fixations during free viewing and finger tapping, they also explored a larger area of the screen as compared to staying more central during fixed viewing (see Fig. 2). There was also a main effect of memory load (F(2, 34) = 9.92, p < .001, ηp2=.368, BF10 = 7.155) on the number of fixations, with a decreasing linear trend (F(1, 17) = 10.92, p = .004, ηp2=.391) across the 0-back (M = 144.56, SE = 10.90), 1-back (M = 146.52, SE = 10.36), and 2-back (M = 128.24, SE = 10.27) conditions. These main effects were qualified by a significant interaction of viewing condition with n-back load on fixation count (F(4, 68) = 3.52, p = .011, ηp2=.171, BF10 = 1.360; see Fig. 3A). Post hoc tests of simple effects revealed that whereas participants did show a linear decrease in fixations with successive memory loads for free viewing (F(1, 17) = 14.21, p = .002, ηp2=.455), they did not demonstrate a significant linear effect of load for the fixed viewing (F(1, 17) = 4.16, p = .057) or finger tapping (F(1, 17) = 0.19, p = .667) conditions.

Figure 2 Fixation Heat Maps.

Heat maps depicting where on the computer screen fixations occurred in each of our three viewing conditions—free (A) and fixed (B) viewing, and finger tapping (C). The heat map for each condition has been averaged across the three n-back loads—0-, 1-, and 2-back—as well as across all 18 participants included in the final analysis. The x and y coordinates reflect the dimensions of the computer monitor used during testing—1,000 × 800 pixels. The warmer colours correspond to locations on the screen where greater numbers of fixations were concentrated, whereas the cooler colours show locations with fewer fixations. Thus, this figure demonstrates that in general, participants complied with our viewing instructions; they explored a larger portion of the screen during the unconstrained eye movement conditions—free viewing and finger tapping—while restricting their gaze toward the central fixation cross during fixed viewing.

Figure 3 (A) Number of fixations and (B) average saccade amplitude (ASA; measured in degrees of visual angle) made by participants on the 0-, 1-, and 2-back tasks during the free viewing, fixed viewing, and finger tapping conditions.

In both plots, individual data points (0-back in red, 1-back in green, 2-back in blue) and means (represented by horizontal bars) are shown for each condition. Error bars represent standard error of the mean (*p < .05). Data for each viewing condition are shown in separate grids, from left to right. (A) Participants made significantly more fixations during free viewing and finger tapping than during fixed viewing. There was also a significant linear decrease in fixations with increasing memory load conditions (0-, 1-, and 2-back). These main effects were qualified by a significant interaction, whereby the linear decrease in fixations with increasing load was observed during free viewing, but not during either fixed viewing or finger tapping. (B) Participants had significantly higher ASA values during free viewing and finger tapping than during fixed viewing.

As for ASA, we observed the expected main effect of viewing condition on saccade amplitude (F(2, 34) = 42.83, p < .001, ηp2=.716, BF10 = ∞). In keeping with the task instructions, participants executed larger amplitude saccades during the conditions allowing free examination of the screen—free viewing (M = 3.78, SE = 0.32) and finger tapping (M = 3.60, SE = 0.38) –than during the fixed viewing (M = 1.36, SE = 0.12) condition (vs. free viewing: F(1, 17) = 66.86, p < .001, ηp2=.797, BF10 = 1.31114; vs. finger tapping: F(1, 17) = 39.32, p < .001, ηp2=.698, BF10 = 5.0189). There was no main effect of memory load on ASA (F(2, 34) = 0.45, p = .643, BF10 = 0.064), nor was there an interaction of viewing condition with n-back load (F(4, 68) = 0.30, p = .878, BF10 = 0.020; see Fig. 3B).

N-back task

Participants differed in their RTs across the three memory loads. As expected, the main effect of memory load on response times was significant (F(2, 28) = 10.18, p < .001, ηp2=.421, BF10 = 645.720), with a linear increase (F(1, 14) = 13.04, p = .003, ηp2=.482) in RT from the 0-back (M = 808.35, SE = 16.43) to 1-back (M = 866.05, SE = 19.30) to 2-back (M = 866.97, SE = 23.46) task across viewing conditions. There was no main effect of viewing condition on RT (F(2, 28) = 1.31, p = .286, BF10 = 0.192), nor was there a significant interaction between viewing condition and memory load (F(4, 56) = 1.20, p = .323, BF10 = 0.169; see Fig. 4A).

Figure 4 (A) Reaction times (RT; measured in milliseconds) and (B) accuracy (measured as d’ scores, which represent z(hits) minus z(false alarms)) on the 0-, 1-, and 2-back tasks during the free viewing, fixed viewing, and finger tapping conditions.

In both plots, individual data points (0-back in red, 1-back in green, 2-back in blue) and means (represented by horizontal bars) are shown for each condition. Error bars represent standard error of the mean (*p < .05). Data for each viewing condition are shown in separate grids, from left to right. (A) There was a significant linear increase in RT with increasing memory load (0-, 1-, and 2-back). (B) There was a significant linear decrease in d’ with increasing memory load (0-, 1-, and 2-back). Participants had lower d’ scores during finger tapping than during either free or fixed viewing.

Considering task accuracy, the expected main effect of working memory load on d’ scores was observed (F(2, 32) = 33.99, p < .001, ηp2=.680, BF10 = 1.9829). When collapsed across the viewing conditions, accuracy decreased with increasing memory load, with d’ scores becoming progressively lower from the 0-back (M = 3.02, SE = 0.10) to 1-back (M = 2.69, SE = 0.13) to 2-back (M = 2.08, SE = 0.13) condition (F(1, 16) = 60.60, p < .001, ηp2=.791). There was also a significant main effect of viewing condition (F(2, 32) = 6.89, p = .003, ηp2=.301, BF10 = 40.970) on task accuracy. Averaged across memory loads, participants had significantly lower d’ scores during finger tapping (M = 2.35, SE = 0.11) than either free (M = 2.83, SE = 0.10) or fixed viewing (M = 2.62, SE = 0.16) (vs. free viewing: F(1, 16) = 15.34, p = .001, ηp2=.489, BF10 = 83.855; vs. fixed viewing: F(1, 16) = 5.07, p = .039, ηp2=.241, BF10 = 1.145). While we were unable to test the free-fixed viewing comparison using a planned orthogonal contrast, we could run a post hoc comparison as part of our complementary Bayesian ANOVA to probe this effect. This Bayesian comparison yielded a Bayes factor of BF10 = 0.762, meaning that the observed data were less likely under the alternative than the null hypothesis, and thus suggesting that there was no difference in task accuracy between the free and fixed viewing conditions. There was no significant interaction of viewing condition and memory load on d’ scores (F(4, 64) = 1.01, p = .408, BF10 = 0.630; see Fig. 4B).

Discussion

Previous studies have shown that free viewing has beneficial effects on long-term memory performance relative to fixed viewing (Henderson, 2003; Johansson et al., 2012; Johansson & Johansson, 2014; Lenoble, Janssen & El Haj, 2018). Although these effects may be related to interference with visual processing due to fixation, the possibility of a general resource reduction due to the top-down attentional control required by the oculomotor fixation task cannot be ruled out as performance on memory tasks in this literature require both visual mnemonic processing and attentional control.

In order to isolate the attentional effects, we used an n-back test of working memory as the primary task. We compared the effects of fixation and a finger tapping task known to interfere with attentional control to free viewing in terms of their interference on an n-back task. Consistent with previous literature (Quinn & Ralston, 1986; Moscovitch, 1994), we found that finger tapping significantly interfered with n-back performance relative to both free and fixed viewing. There was no evidence that fixed viewing interfered with working memory performance as measured by the n-back task, when compared to the free viewing condition, a conclusion supported by Bayesian analysis of the low likelihood of such an effect given the observed data. These findings suggest that the negative impact of sustaining fixation on memory performance, as reported in the broader literature, cannot be accounted for by a general resource reduction due to motor responses secondary to the instructions to fixate the eyes.

To assess adherence to our viewing task instructions, we compared fixation count and ASA across the viewing conditions and memory loads. Participants made more fixations and had higher ASAs during the two conditions that allowed free examination of the screen—free viewing and finger tapping—compared to the fixed viewing condition. These results—combined with the observation of successful finger tapping here and in a previous study using the same pattern (Moscovitch, 1994)—suggest that participants complied with task instructions during each of the tasks. We also observed a linear decrease in fixations with successive memory loads, although this effect was exclusive to free viewing. During free viewing, participants explored the screen when fewer resources were required for the lower load conditions, whereas they reduced the number of fixations (but not the size of their saccades) as task difficulty increased. Eye movements were explicitly restricted by the task instructions during fixed viewing. On the other hand, during finger tapping, fewer resources were available for visual exploration because of the dual task nature of this general motor condition (Pashler, Carrier & Hoffman, 1993). Thus, for different reasons, eye movements were relatively reduced across memory loads—including the less difficult low load condition—during both fixed viewing and finger tapping. We also observed expected main effects of memory load on both RT and accuracy performance (Sternberg, 1966; Jonides et al., 1997; Jensen & Tesche, 2002).

The general finding that finger tapping disrupted verbal working memory is consistent with past research in which arm movements and finger tapping have been shown to disrupt working memory recall of sentences (Quinn & Ralston, 1986), words (Moscovitch, 1994), and nonsense words (Friedman, Polson & Dafoe, 1988). This effect demonstrated that the dual task paradigm employed here is capable of detecting interference from a secondary motor task. As for eye movements, because our fixed viewing condition constrained saccades within a visual window rather than enforcing strict fixation per se, we are unable to draw conclusions about the effect of our working memory task on microsaccade rate, as has been observed in past studies (Siegenthaler et al., 2014; Gao, Yan & Sun, 2015; Dalmaso et al., 2017). That said, we can frame our results in light of previous work showing that controlled eye movements (like our constrained eye movements during fixed viewing) disrupt working memory (Postle et al., 2006). In this study, this disruption was specific to working memory for locations, and did not generalize to visual working memory for shapes (Postle et al., 2006), nor for the verbal working memory task explored here. Altogether, such findings suggest that maintaining sustained fixation does not result in a general decline in attentional resources, as working memory performance would have been disrupted in each of the tasks noted above. Rather, such findings suggest that eye movements may be important for the rehearsal and retrieval of spatial information, consistent with our prior work in which viewers rehearse previously studied spatial locations with their eye movements (Olsen et al., 2014; Wynn et al., 2018) and other findings that describe the ‘looking at nothing’ phenomena (Brandt & Stark, 1997; Laeng & Teodorescu, 2002).

Maintaining fixation likely disrupts visual processes that include the integration and binding of information that would ordinarily occur across eye movements (Liu et al., 2017; Liu et al., 2018) as well as the retrieval or rehearsal of visual details from memory (Brandt & Stark, 1997; Laeng & Teodorescu, 2002; Olsen et al., 2014; Wynn et al., 2018), that is supported through a myriad of connections between the oculomotor and memory systems (Shen et al., 2016; Inman et al., 2017; Ryan et al., 2018). The present results contribute to this larger literature by ruling out general resource effects as a confounding factor to explain the negative impact of sustained fixation on memory performance.

Conclusions

The free versus fixed viewing manipulation is a common method to investigate the effects of eye movements on mnemonic tasks. There is ambiguity related to the origin of such effects, with one possibility being that they are due to a general resource reduction owing to the top-down motor effects of fixating the eyes. Our findings are incompatible with such an account. Although the findings of this study do not delineate the precise mechanisms of the fixation effect on memory, by ruling out a general resource reduction account, they build on prior accounts that suggest that eye movements are functional for memory through its role in binding and reactivation of visual details (Shen et al., 2016; Liu et al., 2017; Liu et al., 2018; Ryan et al., 2018).

Supplemental Information

Dataset S1 Eye movements and working memory

This dataset includes all measures of interest - (1) eye movement measures (number of fixations, average saccade amplitude), (2) n-back task measures (reaction time, d’).

Click here for additional data file.

Carrie Esopenko assisted with the development of the auditory n-back task. Jessica Luk assisted with recruitment and data collection. Douglas McQuiggan and Kelly Shen assisted with the design and data collection of the eye movement monitoring n-back paradigm. Yushu Wang and Jordana Wynn assisted with data analysis and visualization.

Additional Information and Declarations

Competing Interests

Author Contributions

Ethics

Data Availability

Michael J. Armson was a trainee at Baycrest Health Sciences throughout the completion of this project. Jennifer D. Ryan and Brian Levine are employed by Baycrest Health Sciences.

Michael J. Armson conceived and designed the experiments, performed the experiments, analyzed the data, prepared figures and/or tables, authored or reviewed drafts of the paper, approved the final draft.

Jennifer D. Ryan and Brian Levine conceived and designed the experiments, contributed reagents/materials/analysis tools, authored or reviewed drafts of the paper, approved the final draft.

The following information was supplied relating to ethical approvals (i.e., approving body and any reference numbers):

The study was approved by the Baycrest Research Ethics Board (02-01).

The following information was supplied regarding data availability:

The raw measurements are available in a Supplemental File.

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
