# Peer review of "Maintaining fixation does not increase demands on working memory relative to free viewing"

_PeerJ, doi:10.7717/peerj.6839_

## Round 0.1 · original submission · Major Revisions

Please address the enclosed reviewer comments in the resubmission, especially those concerned with the experimental design and validity of the findings.

Reviewer 1 ·

Basic reporting

The paper is well written overall. In order to increase its readership, the paper would benefit enlarging the focus of the introduction (and discussion of the current results) by including reference to a growing body of evidence showing that the frequency of fixational eye movements during prolonged fixation is strongly influenced by tasks involving working memory to different degrees (e.g., Dalmaso et al., 2007; Gao et al., 2015; Siegenthaler et al., 2014).

Here are the references of the papers:
Dalmaso, M., Castelli, L., Scatturin, P., & Galfano, G. (2017). Working memory load modulates microsaccadic rate. Journal of Vision, 17(3):6, 1–12, doi:10.1167/17.3.6.
Gao, X., Yan, H., & Sun, H.-J. (2015). Modulation of microsaccade rate by task difficulty revealed through between- and within-trial comparisons. Journal of Vision, 15(3):3, 1–15, doi:10.1167/15.3.3.
Siegenthaler, E., Costela, F. M., McCamy, M. B., Di Stasi, L. L., Otero-Millan, J., Sonderegger, A., . . .Martinez-Conde, S. (2014). Task difficulty in mental arithmetic affects microsaccadic rates and magnitudes. European Journal of Neuroscience, 39, 287–294.

In the first sentence of the abstract, I would suggest deleting "eye movements and". I think the sentence is misleading as it stands. The comparison is between memory performance under different viewing conditions, not between eye movements.

Experimental design

The question addressed in the manuscript is well defined and important for clarifying different accounts in the literature. The dual task methodology adopted in the experiment seems appropriate and the overall methods are sound. That said, my first point is that the inclusion of a secondary task (finger tapping, see text in lines 92-93) able to modulate the auditory n-back task does not eliminate the possibility that a lack of difference in primary task performance as a function of free viewing vs. fixation may reflect a lack of sensitivity of the primary task. In addition, and more critically with respect to the experiment, I am not really convinced that keeping the eyes within a 400x400 window can be considered as strict fixation. This, in turn, casts serious doubts about the possibility that the two eye-related secondary tasks were really different (the presence of a main effect in the number of fixations does not rule out this possibility). The authors should address this issue very carefully. Put it more explicitly, I wonder what would happen to memory performance by using more demanding fixation constraints (e.g., those used in studies on fixational eye movements).

Validity of the findings

The dual task paradigm implemented by the authors is informative but potentially weak within a pure Null Hypothesis Significance Testing framework. I strongly suggest the authors to complement the current analyses with other analyses using a Bayesian approach which can deal with the problem of accepting the null hypothesis (lack of difference in memory performance between free viewing and "fixation") more appropriately (e.g., Masson, 2011; Rouder et al., 2017).
Here are the references of the papers:
Masson, M. E. J. (2011). A tutorial on a practical Bayesian alternative to null-hypothesis significance testing. Behavior Research Methods, 43, 679-690.
Rouder, J. N., Morey, R. D., Verhagen, A. J., Swagman, A. R., & Wagenmakers, E.-J. (2017). Bayesian analysis of factorial designs. Psychological Methods, 22, 304-321.

Reviewer 2 ·

Basic reporting

This manuscript is well-written. Literature review is sufficient. Of general interest may be to include comparisons with the results found in Siegenthaler et al. (2014) where they did find effect on saccade amplitude during a cognitive workload task.

Hypotheses are well addressed.

Figures are suitable but a few changes would make them better:
- Since the 4 figures share the same format, I would merge figures 1 and 2 in the same panel, and figures 3 and 4 in another one, saving space in the figure legends.
- Figure 3 does not show error bars so that text should be removed from the corresponding legend.
- The colors of the dots appear the same when printed in a B&W version, so authors could change the colors of the 3 n-back load tasks.
- Crosses should cover (bring to the front) the cloud of dots and not otherwise.

Experimental design

Methods need additional detail to understand the experimental design - How many practice trials did participant complete? How many blocks in total? How long were the trials? Was there a time limit after showing each letter for the participant to say yes? What happened if participants failed to tap the pattern correctly? and what was the distribution of repeated letters per trials (i.e. what was the percentage of required 'yes' per trial?).

Section 'Computers' does not add to the manuscript and it could be omitted.

Two first sentences in paragraph 'Software' are long and tedious to read. Simplify.

line 193 - Missing right parenthesis. Not clear what 'out of lure' means.

line 209 - Additional space

lines 231-237 - This belongs to Introduction

line 345 - Authors repeat sentence from 342 but replacing "spatial information" with the unclear "object and/or verbal information". Define exactly what object means or remove sentence.

line 361 - Remove 'our' at the end of the line

Validity of the findings

Armson, Ryan, and Levine found no evidence that fixed viewing interfered with working memory performance as measured by the n-back task, when compared to the free viewing condition. These findings fail to support a hypothesis of increased working memory load during fixation but rule out general attentional effects in the process, bringing new insights into the field. The results are conclusive. Data analysis was robust.

Additional comments

Minor edits:
line 81 - "top-Down"
line 99 - This sentence is odd - probably remove "in and of"
line 108 - Approval number is not necessary

Reviewer 3 ·

Basic reporting

Different effects of eye movements in free and fixed viewing conditions on memory encoding and retrieval has been reported. But the mechanism of the memory reduction during fixed viewing is still unclear. The present study using a working memory task – an auditory n-back task. They asked the participants to perform this task during free and fixed viewing, as well as a repetitive finger tapping. The authors reported there was no significant interactions in fixed viewing with n-back performance relative to free viewing, which failed to support a hypothesis of increased working memory load during fixation.

Experimental design

The research question seems timely and the approach reasonable. However, some crucial information is missing from the manuscript that would be necessary to fully assess the soundness of the work. For example, although the auditory n-back task has been used in several previous studies, it is still necessary to demonstrate the experimental floe chart as figure in the manuscript.

Validity of the findings

Additionally, I do not find the authors state explicitly how to measure of the number of fixations. As they reported, even during the fixed viewing condition, the number of fixations was still about 113. My main concern is that the authors did not analyze and interpret their results in depth. Since there was no requirement of eye movement during the free viewing condition, participants might still fixated at the corner of the monitor while they were doing the main n-back task.

Additional comments

Whilst not being a native English speaker, I still found some grammatical issues, typos etc., and therefore suggests additional efforts in language editing. Some specific typos to be modified:
- P 1, Line 37: double ‘in’
- p 6, Line 247: 11.94
- P 9, Line 362: needless ‘the’

Overall, the manuscript is well-written, the Introduction and Discussion sections are sound. It is of potential interest to readers in this field, but I cannot recommend publication in its current form. Moderate revisions are necessary, in my opinion.

---

## Round 0.2 · accepted · Accept

Congratulations and thank you for submitting to PeerJ!

# Reviewer 1 ·

Basic reporting

no comment

Experimental design

no comment

Validity of the findings

no comment

Additional comments

The authors have done a very nice job in addressing my previous concerns. I have no further comments.

Reviewer 2 ·

Basic reporting

This manuscript is well-written. Literature review has been improved. Figures have been successfully modified as requested.

Experimental design

Hypotheses are well addressed. Methods are easier to read now.

Validity of the findings

No comment

Additional comments

The authors have successfully addressed my concerns.

Reviewer 3 ·

Basic reporting

no comment

Experimental design

no comment

Validity of the findings

no comment

Additional comments

The authors have largely improved their manuscript. This is now a clear and interesting paper. I recommend publication in its current form.